# The Design and Preparation of Permittivity-Adjustable FeNi@SrFe-MOF Composite Powders

**Zhifen Yuan** [1], **Donghan Jiang** [1], **Lei Chen** [2] **and Zhenghou Zhu** [1,*]

1   School of Physics and Material Science, Nanchang University, Nanchang 330031, China
2   Jiangxi Hongdu Aviation Industry Group Co., Ltd., Nanchang 330024, China
*   Correspondence: zhuzhenghou@ncu.edu.cn

**Abstract:** When the thickness of the wave-absorbing material is low, there exists the problem of the narrow wave-absorbing frequency band, making it difficult to regulate the position of the wave-absorbing peak. In this study, FeNi@SrFe-MOF composite powders were synthesized using a hydrothermal method and a liquid-phase reduction method. The composite powder was spherical, with a particle size of about 50 μm–60 μm; the core layers of the powders were porous SrFe-MOF powders with permanent magnetization, and the outer layers were FeNi alloy nano-powder coatings with a particle size of 100 nm–120 nm, which took into account both the soft magnetization and the permanent magnetization properties of the composite powders. Additionally, a directional magnetic field was applied to the powder coating. By regulating the intensity and direction of the magnetic field, the electromagnetic parameters of the composite powder coating underwent sensitive changes, allowing for the precise regulation of the electromagnetic wave absorption performance of the composite powders. With the increase in magnetic field intensity, the $\varepsilon'$ value decreased significantly. The $\varepsilon'$ values were 8.56–7.35 for H453mT and 6.73–6.12 for H472mT. When no magnetic field was applied, the Snoke limit frequency of the $\mu'$ value was 6.0 GHz. When the magnetic field intensity increased, the Snoke limit frequency of the $\mu'$ value increased from 6.0 GHz, without the magnetic field, to 8.3 GHz; the Snoke limit of the composite powders was shattered. After the H453mT magnetic field regulation treatment, the powder coating exhibited good impedance matching characteristics with air. When the magnetic field intensity was 453mT and the thickness of the composite powders coating was 3.5 mm, the composite powders coating showed the strongest absorption peak when the R-value was −59 dB at 7.8 GHz, and the effective absorption bandwidth reached 3.2 GHz, exhibiting superb absorbent qualities. The wave absorption property of the coating can be sensitively changed by the magnetic field regulation treatment at the condition without changing the powder structure or coating structure, which provides a new strategy for the regulation of the wave absorption property and has broad application prospects.

**Keywords:** wave-absorbing materials; composite powders; magnetic regulation

## 1. Introduction

Magnetic powder materials, such as the magnetic metal powder, ferrite, are currently the most efficient and commonly used radar-absorbing materials (RAM) [1–11]. Zhu et al. [12] developed those FeNi alloy nano-powders, such as $Fe_{20}Ni_{80}$ and $Fe_{50}Ni_{50}$, with excellent soft magnetic properties, with particle sizes ranging from 15 nm to 150 nm. The coating of $Fe_{50}Ni_{50}$ alloy powders, which has a thickness of 1.5 mm and a frequency within the range of 1 GHz to 18 GHz, has an effective absorption bandwidth of 3 GHz, with the reflection coefficient, |R|, being over 10 dB, exhibiting excellent radar-absorbing properties.

The biggest challenge for magnetic powder absorbing materials is that once the thickness of the powder coating is determined, the absorption peak position is also fixed, and the effective absorption bandwidth is lower, which leads to a poor overall absorbing performance. The ideal properties for RAM, such as the broadband-absorbing feature, lower

thickness, and lower density, are required. The broadband-absorbing feature means that the effective absorption bandwidth is over 10 GHz when the frequency ranges from 1 GHz to 18 GHz. It is very difficult to achieve the broadband-absorbing feature.

In response to the current absorbing requirements of the broadband-absorbing feature, lower thickness, and lower density, Liu et al. [13] constructed new kinds of $FeNi@C@Fe_3O_4$ composite powders, which combine the characteristics of dielectric loss- and magnetic loss-absorbing agents. The composite powders had the effective absorption bandwidth of more than 5 GHz and the lower density, $\rho$, of 1.75 g/cm$^3$, which is a significant advantage. However, the absorption peak of the powder is located at 8 GHz, and there is no absorption peak in the C-band below 8 GHz, which shows that the wide-band absorbing property cannot be achieved.

To address the current absorbing requirements of the broadband-absorbing feature, lower thickness, and lower density for RAM, inspired by the magnetic regulation robot [14] and related magnetic field regulation articles [15–20], this study constructed new kinds of FeNi@SrFe-MOF composite powders (FSM powders). These powders fully leverage the lightweight properties of MOFs and the excellent soft magnetic properties of FeNi alloy nano-powders to solve the problem of the poor magnetic absorbing property in the case of the lower material thickness below 5 mm and the frequency below 8 GHz. What is more, the important innovation is that the center of the FSM particle is the $SrFe_{12}O_{19}$ particle, with permanent magnetic properties. When FSM powders are exposed to an external magnetic field, those powders can respond quickly to cause some changes in the composite powders, such as internal stress, which causes a change in the electromagnetic parameters of FSM. The absorbing characteristics of the composite powders also change significantly with the change in the electromagnetic parameters, which can achieve the goal of effectively controlling the absorbing property. To address the issue of poor absorbing property when the material thickness is below 5 mm and the frequency band is below 8 GHz, this study selected a research object that had a material thickness below 5 mm and a frequency band ranging from 1~8.5 GHz.

## 2. Experiment

The microstructures of the FSM powders were observed using field emission scanning electron microscopy (Quanta 200FEG, FEI, USA) and an orthographic metallographic microscope (RX50M, Ningbo Shunyu Instrument Co., Ltd., Ningbo, China). The phase compositions of the FSM powders were detected by X-ray diffraction (D8 ADVANCE, Bruker, Billerica, MA, USA). The vector network analyzer (Agilent PNA N5244A, Santa Clara, CA, USA) and the Agilent E5063A material electromagnetic property test analyzer were used to test the electromagnetic parameters of FSM powders in the microwave frequency band of 1 GHz~8.5 GHz.

### 2.1. Preparation of FSM Powders

The $SrFe_{12}O_{19}$ powders with a particle size of about 5 μm were dispersed in liquid butylbenzene rubber. The liquid rubber was heated to 150 °C for 12 h in an $N_2$ atmosphere and then was calcined at 300 °C for 2 h to generate the black SrFe-MOF powders. The powders were dispersed into the aqueous solution of $FeSO_4 \cdot 7H_2O$ and $NiSO_4 \cdot 6H_2O$, the temperature was maintained at 85 °C for 1 h, and the PH value was more than 12. The $NH_3$ gas was continuously generated during the reaction process, and finally, the black powders were generated in the water. The final FSM powders were obtained by magnetic separation and drying. The element ratio of Fe and Ni in the FSM composite powders was 50:50. Figure 1 shows the preparation process of the FSM powders.

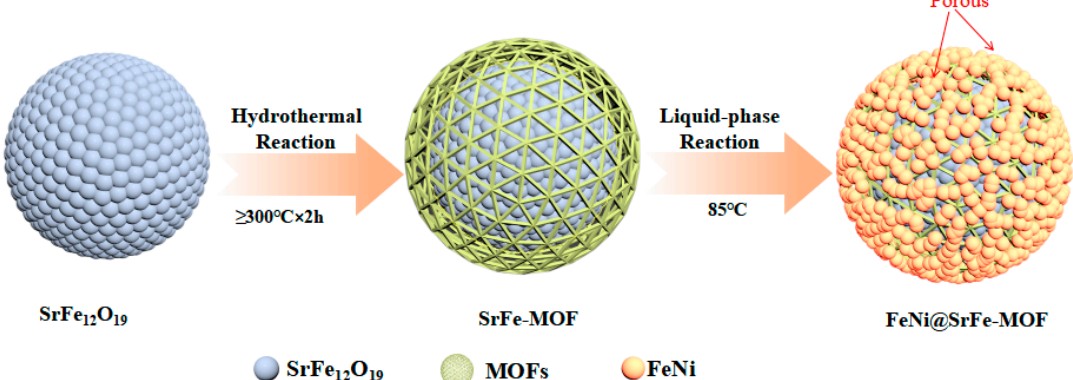

**Figure 1.** Schematic diagram of the composite powder formation.

*2.2. Magnetic Regulation of Electromagnetic Parameters*

The FSM powders were magnetized under a 1.0 T pulsed DC magnetic field. After magnetization, the FSM powders were mixed with flexible epoxy resin (E51) in a ratio of 8.5:1.5 to prepare the coaxial sample of the FSM/E51-absorbing composite material. The outer diameter of the coaxial sample was 7.00 mm, and the inner diameter was 3.04 mm. The electromagnetic parameters of the sample were obtained by a vector network analyzer (Agilent PNA N5244A) using coaxial line testing. The density of the absorbing material was adjusted by an external DC magnetic field. When the magnetic pole direction of the external DC magnetic field was the same as that of the absorbing material magnetic pole, the absorbing material was compressed; otherwise, it was stretched. By changing the direction of the external magnetic field, the compression and stretching of the FSM/E51-absorbing composite material can be conveniently adjusted. The schematic diagram of magnetic regulation is shown in Figure 2.

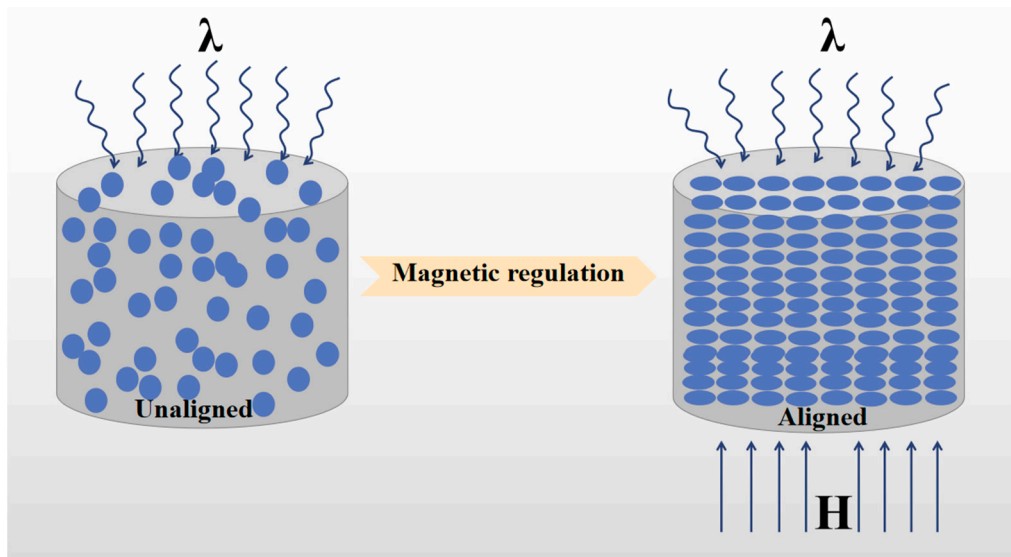

**Figure 2.** Magnetic regulation schematic.

## 3. Morphology and Structure of Composite Powders

As observed by SEM, the FSM powders were black powders with a particle size of about 50 μm–60 μm, the MOF surface adsorbed a large number of FeNi alloy nano-powders with particle sizes of about 100 nm–120 nm, and the nano-powders piled up into a multilayer structure (as shown in the a-area in Figure 3a), while there were bare parts on the localized surface of MOF (as shown in the b-area in Figure 3a). The porous structure on the surface of the FSM powders was due to the porous structure on the surface of MOF and the pileup of multilayer powders on the surface; at the same time, a large number of FeNi alloy

nano-powders were also present in the porous interstices. The EDS (Figure 4) analysis of the FSM powders showed that the main composition of the part covered by the FeNi alloy nano-powder included Fe and Ni elements, and the composition of the bare part of the MOF surface included Fe, Sr, O, and C elements. This analysis showed that the structure of the FSM powder was as follows: the outer layer was the FeNi alloy nano-coating layer, and the core layer was the SrFeO core, with permanent magnetic properties prepared into porous SrFe-MOF powders, which finally formed the FSM composite powders. The powder formed a core–shell structure (as shown in Figure 5), due to the combination of the meso-FeNi alloy nano-powder and the SrFe-MOF powder, by magnetic force.

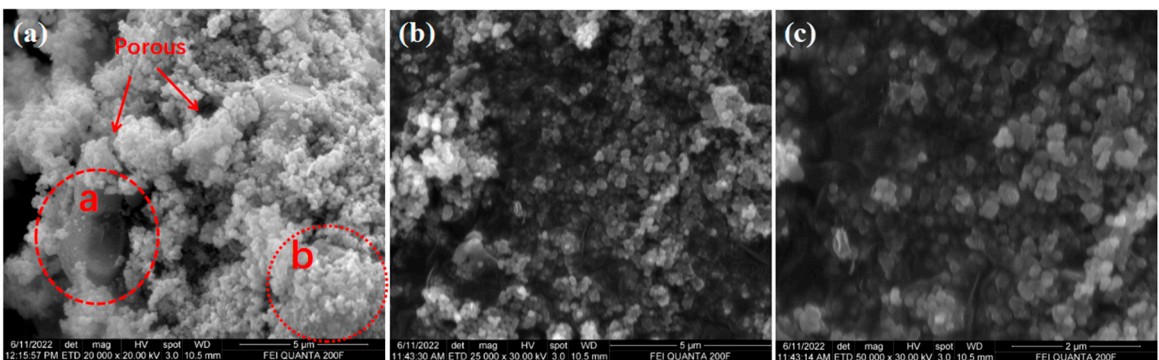

**Figure 3.** (**a**–**c**) SEM images of FSM composite powders.

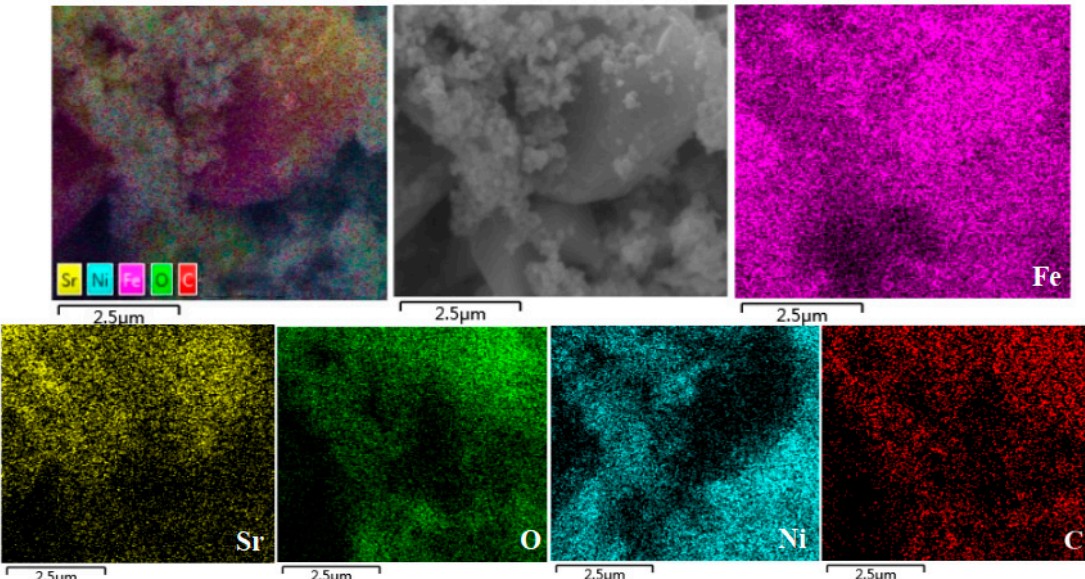

**Figure 4.** EDS images of FSM composite powders.

The main phase structure of the FSM powder consisted of the $FeNi_3$ phase at the surface sites, amorphous carbon in the intermediate layer, and the $SrFe_{12}O_{19}$ phase at the core sites. The XRD pattern of the FSM showed the presences of $FeNi_3$ and $Fe_2O_3$ phases in addition to the diffraction peaks of the SrFe-MOF phase. Diffraction peaks of the $FeNi_3$ phase at the 2θ angles of 44.2°, 51.5°, and 75.8° corresponded to the (111), (200), and (220) crystal planes. Those diffraction peaks at 2θ of 35.4° and 53.4° corresponded to the (311) and (422) crystal planes, representing the $Fe_2O_3$ phase. Therefore, EDS analysis showed that the main elements in the FSM powders were Fe, Ni, O, C, and Sr (Figure 6b), which was consistent with the XRD analysis. The strong absorption peak at 584 cm$^{-1}$ indicated the absorption peak of the Fe-O bond vibration or the Sr-O bond vibration; 758 cm$^{-1}$ was the absorption peak of magnetic lattice vibration in strontium ferrite; the absorption peak of the stretching vibration of C-H occurred at 2868 cm$^{-1}$; and the absorption peak at 3702 cm$^{-1}$

was the stretching vibration associated with the hydrogen on the surface of iron, suggesting that the FSM powders were successfully prepared (Figure 6c).

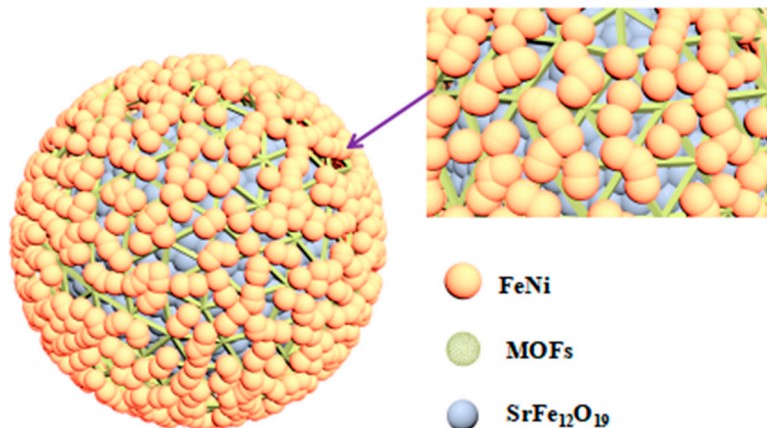

**Figure 5.** Structure of the FSM composite powders.

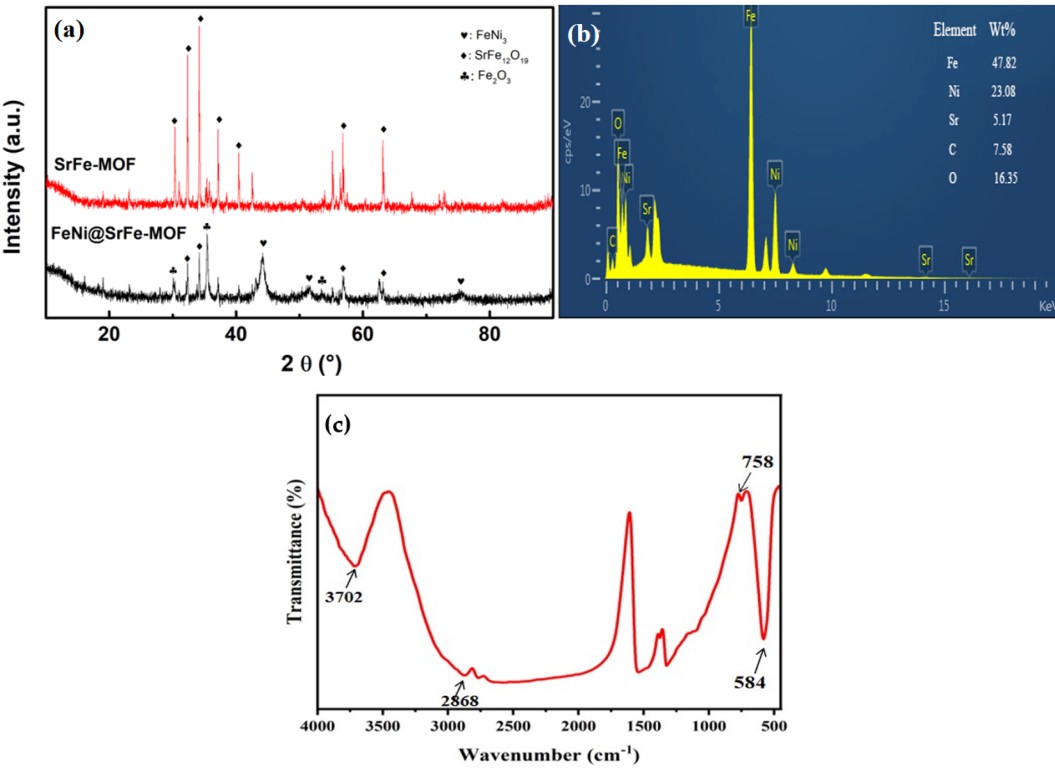

**Figure 6.** (**a**) XRD images of FSM composite powders, (**b**) EDS images of FSM composite powders, and (**c**) the FTIR of FSM composite powders.

The liquid-phase reduction reaction method was used to prepare FeNi alloy nano-powders. The FeNi alloy nanoparticles on the surfaces of FSM powders had a uniform particle size distribution of about 100 nm–120 nm and were spherical (Figure 7). The optimized process for preparing FeNi alloy nano-powders was as follows: $[N_2H_4]/([Ni^{2+}] + [Fe^{2+}])$ was 2/1, $[Ni^{2+}]/[Fe^{2+}]$ was greater than 1/1 (molar ratio), the pH value was 13, the reaction temperature was more than 80 °C, and the reaction time was greater than 30 min. The process of the reaction is as follows:

$$FeSO_4 + NiSO_4 + N_2H_4 + 4NaOH \rightarrow FeNi \downarrow + N_2 \uparrow + 4H_2O + 2Na_2SO_4 \qquad (1)$$

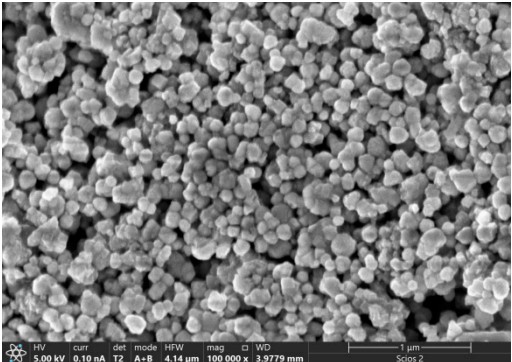 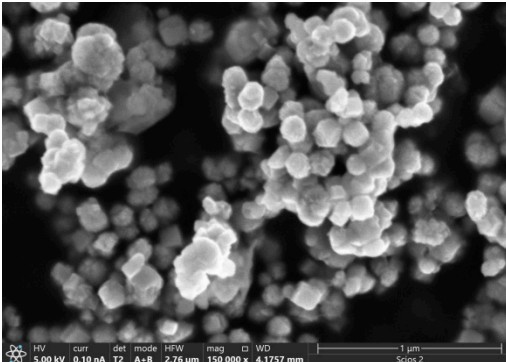

**Figure 7.** Morphologies of FeNi alloy nano-powders.

## 4. Electromagnetic Properties and Absorption Performances of Composite Powders

FSM powders exhibited good soft magnetic properties and certain conductivity characteristics. However, after FSM powders were combined with epoxy resin, the permeability μ and permittivity ε of the FSM/epoxy resin composite materials did not reach the percolation threshold, so the permeability μ and permittivity ε of the composite material were still not higher. The electromagnetic wave loss of the FSM powders depended mainly on the magnetic loss mechanism, and the tangent of the magnetic loss tgδμ value was 0.23~0.44 (Figure 8f). Figure 9 is a schematic diagram of the possible electromagnetic wave absorption mechanism of the composite powder.

FSM powders primarily exhibit magnetic loss mechanisms. The eddy current loss effect is the main contributor to magnetic loss mechanisms [21].

$$C_0 = (\mu''(\mu')^{1/2})/f = 2\pi\mu_0\sigma d^{2/3} \tag{2}$$

where $C_0$ is the eddy current loss effect coefficient, μ0 is the vacuum permeability, f is the electromagnetic wave frequency, σ is the material conductivity, and d is the material thickness. According to this equation, when the material thickness d is determined, the value of C0 is only related to the material conductivity σ, and the value of σ varies slightly with the frequency f. Therefore, if the magnetic loss is caused by the eddy current effect, $C_0$ will be constant. The $C_0$ of the FSM powders fluctuates with the frequency from 1 to 8.5 GHz. Therefore, the magnetic loss mechanism in the FSM powders includes not only the eddy current loss effect but also the natural resonance loss.

The external regulation magnetic field had a significant impact on the electromagnetic parameters of FSM powders. Without applying the regulation magnetic field, the real part of the permittivity $\varepsilon'$ value ($\varepsilon'$ value) of FSM powders in the frequency range of 1–8.5 GHz was 9.27–8.26 (as shown in Figure 8a). With the increase in magnetic field intensity, the $\varepsilon'$ value decreased significantly. The $\varepsilon'$ value was 8.56–7.35 for H453mT and 6.73–6.12 for H472mT. As shown in Figure 8c, without applying the regulation magnetic field, the real part of the permeability of the $\mu'$ value ($\mu'$ value) of FSM powders decreased from 1.6 to 1.0 in the frequency range of 1–6.0 GHz, which indicates that the Snoke limit frequency of the $\mu'$ value was 6.0 GHz. With the increase in magnetic field intensity, the $\mu'$ value increased significantly. At H472mT and in the frequency range of 1–8.3 GHz, the $\mu'$ value was 1.6–1.0, and the Snoke limit frequency of the $\mu'$ value was 8.3 GHz. Therefore, increasing the regulation magnetic field can break the Snoke limit of the $\mu'$ value of magnetic powders.

The magnetic field regulation of FSM powders is based on the percolation mechanism. In FSM powder samples, particle spacing decreased with the increase in FeNi alloy powder content. When the spacing decreased to a certain value, d0, the magnetic fields around the magnetic particles overlapped each other, and the magnetic permeability of the composite powder increased significantly, which is the magnetic percolation effect of the composite material. The real part of the permeability of the $\mu'$ value of FeNi powder composites increased slowly with the increase in powder content when the powder content was

lower than the percolation value. Figure 10 shows that the density of the FSM powder arrangements increased with the enhancement of the regulatory magnetic field. When the densities of the powders were lower than the percolation threshold, the $\mu'$ value of the powders obeyed the mixing rule of the composite materials. Because of the good magnetic properties of the $Fe_{50}Ni_{50}$ powders, the $\mu'$ value of the composite materials increased with the increase in the density. The $\varepsilon'$ value of the powders followed the mixing rule of the composite materials when the densities of the powders were lower than the percolation threshold value. Because the electrical conductivity of the FSM powders was not good, the electrical conductivity of the composite material decreased with the increase in the density, and the $\varepsilon'$ value decreased significantly.

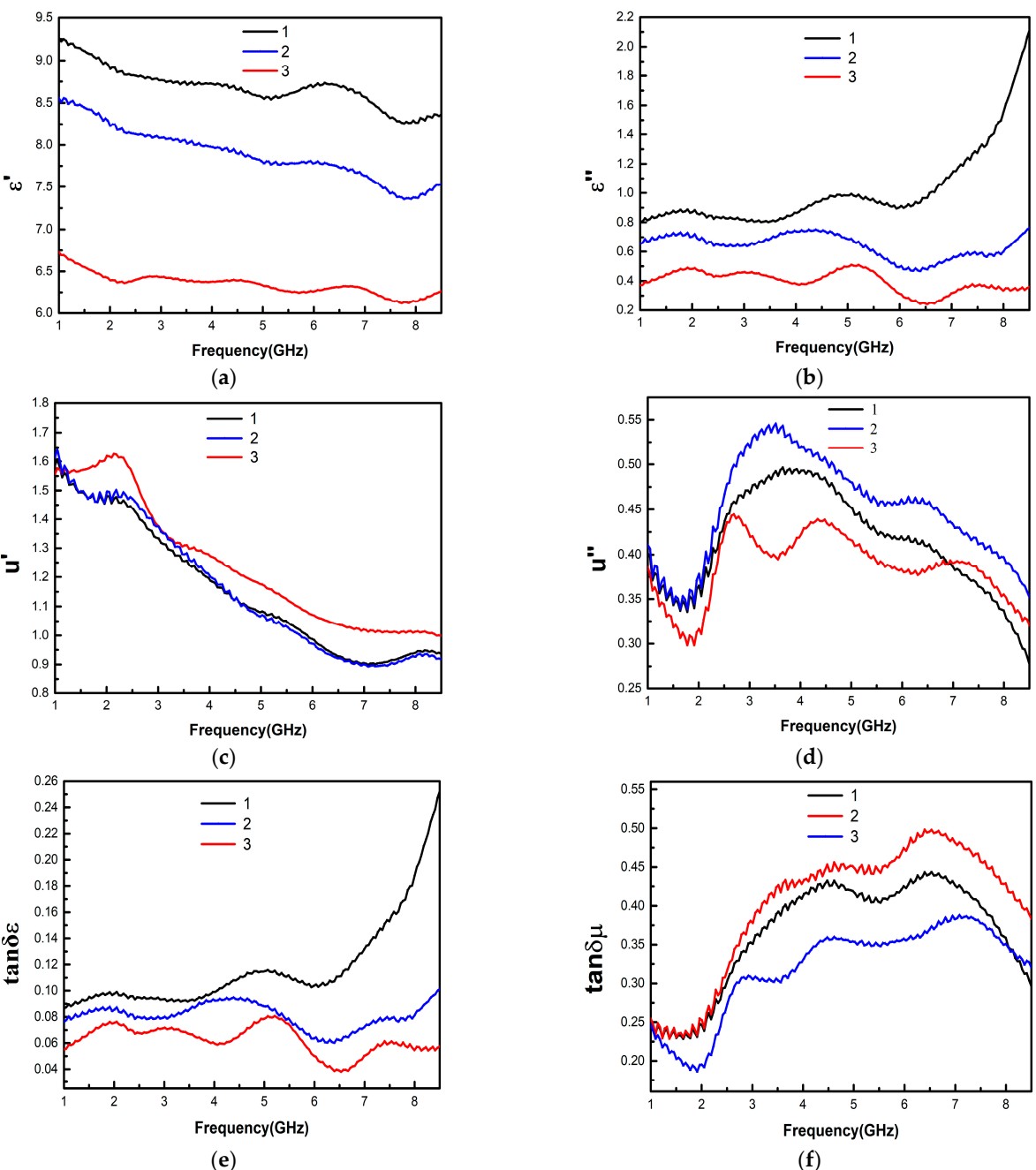

**Figure 8.** (**a**–**f**) Electromagnetic parameters of the composite powders (1—H0; 2—H453mT; 3—H472mT).

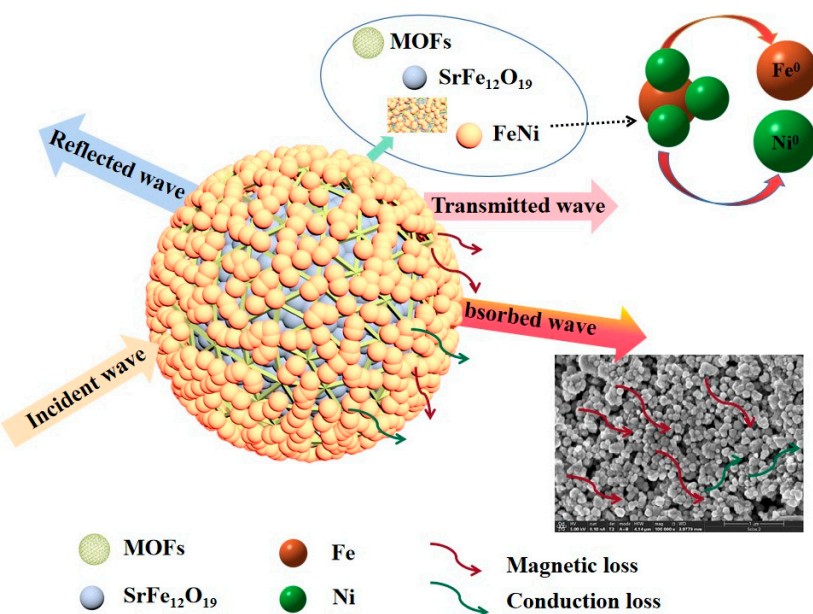

**Figure 9.** Schematic diagram of the possible electromagnetic wave-absorption mechanism of the composite powders.

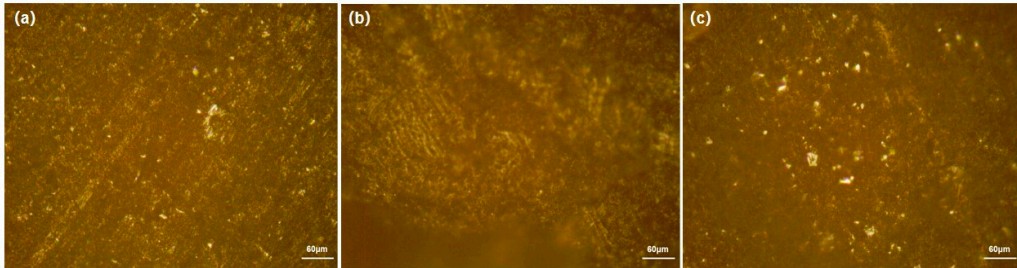

**Figure 10.** Morphologies of the composite powders; ((**a**): H0, (**b**): H453mT, (**c**): H472mT).

In addition to the effect of percolation, there was a significant constraint on the $\mu'$ value of the FSM powders, namely the Snoek effect. The $\mu'$ value was limited by the Snoek limit. There was a Snoek theorem between the initial static permeability $\mu_i$ and the resonance frequency fr:

$$(\mu_i - 1)\text{fr} = \frac{4}{3}\gamma\text{Ms} \tag{3}$$

In Equation (3), $\gamma$ and Ms are the gyromagnetic ratio and the saturation magnetization, respectively. A lower Ms results in a lower $\mu_i$, according to Snoek's theorem. The Ms value of the metal powder FeNi was 62 emu/g [22]. After the FeNi powders were mixed with paraffin wax, the $\mu'$ values of FeNi and paraffin increased, and the material was found to have weak soft magnetic properties.

According to the impedance matching theory, the effective absorption of the incident microwaves requires not only sufficient electromagnetic attenuation capability but also good impedance matching. The surface reflection coefficient calculation formula is as follows:

$$R = (\mu_r \cdot \mu_0/\varepsilon_r \cdot \varepsilon_0 - \mu_0/\varepsilon_0)/(\mu_r \cdot \mu_0/\varepsilon_r \cdot \varepsilon_0 + \mu_0/\varepsilon_0) = (Z - 1)/(Z + 1) \tag{4}$$

where $\varepsilon_0$, $\mu_0$, $E_r$, and $\mu_r$ are the vacuum permittivity, vacuum permeability constant, the complex permittivity, and permeability constants, respectively.

The $(\mu/\varepsilon)^{1/2}$ value was used to characterize the impedance matching, the $(\mu/\varepsilon)^{1/2}$ value of FSM powders was 0.34~0.42 without external magnetic field regulation, and the

impedance matching characteristics were poor in the range of 1~8.5 GHz. After magnetic field regulation, the $(\mu/\varepsilon)^{1/2}$ value of the powders increased significantly, due to the increase in permeability and the decrease in permittivity. The $(\mu/\varepsilon)^{1/2}$ value at H453 mT was 0.36–0.45, and the $(\mu/\varepsilon)^{1/2}$ value at H472 mT was 0.41–0.51. Therefore, after magnetic field regulation, the impedance matching between FSM powders and air was improved, which is beneficial for electromagnetic waves to penetrate the powders and to lay the foundation for the material's interference absorption and loss absorption.

The absorption property of the powders in the frequency range of 1 GHz to 8.5 GHz can be evaluated using the reflection coefficient loss value, R. According to the transmission line theory [23]:

$$R = 20 \lg | (Z_{in} - Z_0)/((Z_{in} + Z_0)) | \tag{5}$$

$$Z_{in} = Z_0 (\mu_r/\varepsilon_r)^{1/2} \tanh ((j (2\pi fd/c)(\mu_r \varepsilon_r)^{1/2}) \tag{6}$$

where d is the thickness of the absorbing coating, $Z_0$ is the characteristic impedance of free space, Zn is the input impedance of the absorbing coating, $\varepsilon_r$ and $\mu_r$ represent the complex dielectric constant and complex magnetic permeability constant, respectively, f is the frequency of the electromagnetic wave, and c is the velocity of the electromagnetic wave in free space. Generally, when the reflection loss R is less than −10 dB, it means that 90% of the incident electromagnetic wave can be lost, and it is considered effective absorption (Figure 11). The frequency band with an R less than −10 dB is considered the effective absorption band (EAB).

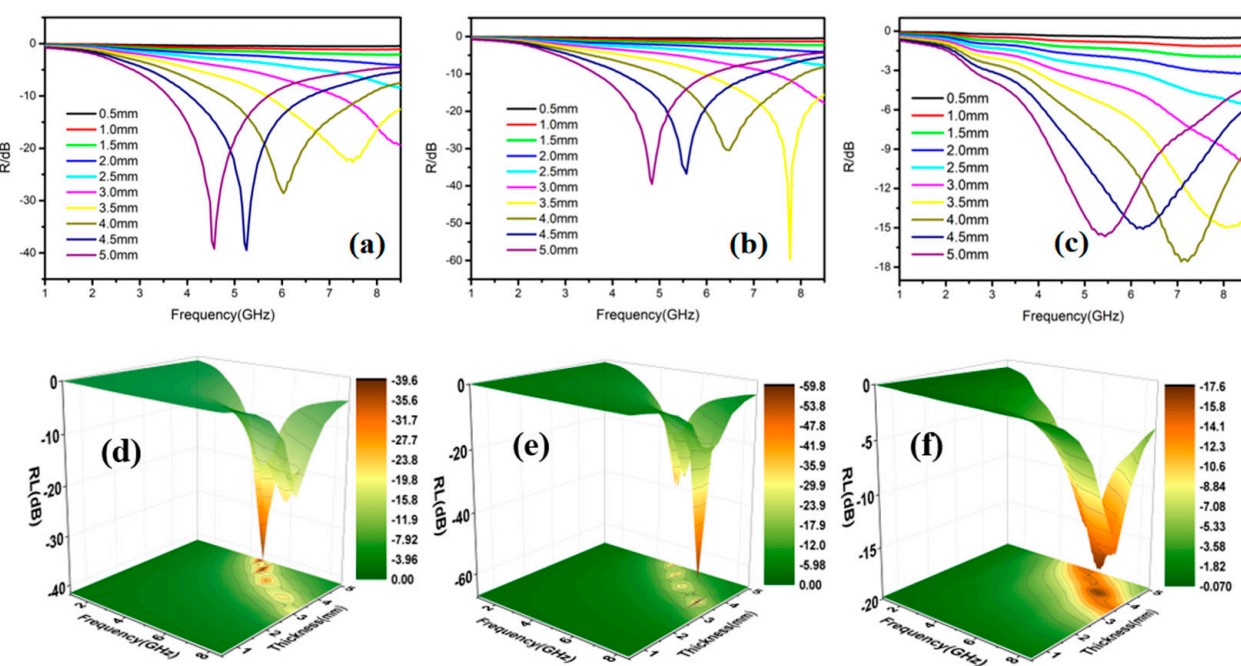

**Figure 11.** R values of the powders with different thicknesses; (**a,d**): H0, (**b,e**): H453mT, (**c,f**): H472mT.

The thickness and absorption band of composite powders under different magnetic fields are shown in Table 1. The $(\mu/\varepsilon)^{1/2}$ value of FSM powders in the band of 1.0 GHz~8.5 GHz is about 0.4. The powder coating exhibits good impedance matching characteristics with air. When the thickness of the powder coating is 4.4 mm, the strongest absorption peak is −41 dB at the center frequency of 5.4 GHz, and the EAB value is 2.7 GHz (Figure 12a). When the thickness of the powder coatings is 3.8 mm, the widest effective absorption peak appears in the frequency band of 5.3 GHz~8.5 GHz, and the EAB value is 3.2 GHz, showing excellent absorption wave properties (Figure 12b).

**Table 1.** Thicknesses and absorption bands of composite powders under different magnetic fields.

| d/mm | H0 | | H453mT | | H472mT | |
|---|---|---|---|---|---|---|
| | Effective Absorption Frequency Band/Bandwidth | Main Absorption Peak Frequency/ Peak (dB) | Effective Absorption Frequency Band/Bandwidth | Main Absorption Peak Frequency/ Peak (dB) | Effective Absorption Frequency Band/Bandwidth | Main Absorption Peak Frequency/ Peak (dB) |
| 3 | 7.13~8.5/1.37 | 8.50/−19.4 | 7.40~8.5/1.1 | 8.50/−17.78 | 8.41~8.5/0.09 | 8.50/−10.26 |
| 3.5 | 5.71~8.50/2.79 | 7.49/−22.66 | 5.94~8.50/2.56 | 7.77/−59.3 | 6.81~8.50/1.69 | 8.04/−15.00 |
| 4 | 4.75~7.72/2.97 | 6.03/−28.61 | 5.02~8.04/3.02 | 6.49/−30.43 | 5.98~8.27/2.29 | 7.08/−17.57 |
| 4.5 | 4.06~6.63/2.57 | 5.25/−39.46 | 4.20~7.08/2.88 | 5.57/−36.78 | 5.02~7.49/2.47 | 6.26/−15.08 |
| 5 | 3.61~5.71/2.1 | 4.57/−39.2 | 3.65~6.08/2.42 | 4.84/−39.54 | 4.43~6.44/2.01 | 5.44/−15.65 |

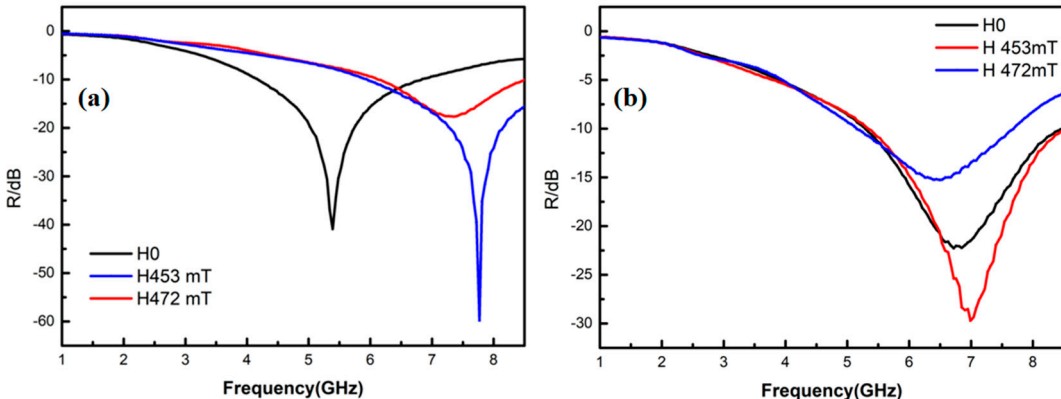

**Figure 12.** Absorption performances of composite powder coatings; ((**a**): strongest absorption peak, (**b**): widest absorption peak).

After the H453 mT magnetic field regulation treatment, the powder coating exhibits good impedance matching characteristics with air. When the thickness of the powder coating is 3.5 mm, the strongest absorption peak at the center frequency of 7.8 GHz is −59 dB, and the EAB value is 2.6 GHz. When the thickness of the composite powder coating is 3.8 mm, the widest effective absorption peak appears in the frequency band of 5.3 GHz~8.5 GHz, and the EAB value is 3.2 GHz. Compared with the case without magnetic field regulation, the coating thickness with the strongest absorption peak decreases from 4.4 mm to 3.5 mm, and the center frequency of the strongest absorption peak changes from 5.4 GHz to 7.8 GHz. The strongest absorption peak is reduced from −41 dB to −59 dB. Similarly, the wave absorption property of the FSM powders is significantly changed after H472 mT magnetic field regulation treatment.

When there is electromagnetic wave radiation to the material surface, electromagnetic wave transmission, reflection, and absorption behavior will occur. Here, the electromagnetic wave absorption loss performance mainly depends on the tgδ value of the material, the intensity of the incident wave entering the material, and the thickness of the material. When the thickness of the material d value is very small, the absorption loss is very low; for the radar wave absorption loss, an important loss mechanism is the interference loss mechanism between the incident wave and the reflected wave. The wavelength of the incident wave will change after entering the material; when the thickness of the material is an integer multiple of 1/4 of the wavelength of the incident wave in the material, the interference of the reflected wave will occur in the front and back interfaces of the material, which will make the reflected wave energy drop significantly. Interference loss is the main mechanism of stealth materials. Interference loss mainly depends on the material electromagnetic parameters, material thickness, electromagnetic wave frequency, and other parameters of the integrated matching. When the material thickness d is small, changes in the electromagnetic parameters are the most likely measures. For the problem of material

electromagnetic parameter adjustment subject to the Snoke limitation and the limitation of material magnetic properties, the magnetic permeability of the material in the GHz frequency band changes in a small magnitude, which is difficult to regulate, while the dielectric constant varies greatly with the material electrical conductivity, so the regulation of the material permittivity is a means that can be realized. Therefore, after magnetic field regulation treatment, the electromagnetic parameters of the FSM powders change significantly, which results in significant changes in the absorption of the coating. Without changing the structure of the powders and the structure of the coating, the magnetic field regulation treatment has the function of sensitively changing the absorption of the wave.

## 5. Conclusions

FeNi@SrFe-MOF powders were synthesized using the hydrothermal method and the liquid-phase reduction method. The composite powder was spherical, with a particle size of about 50 μm–60 μm, the core layer of the powders were porous SrFe-MOF powders with permanent magnetization, and the outer layers were FeNi alloy nano-powder coatings with a particle size of 100 nm–120 nm.

The electromagnetic wave loss of the FeNi@SrFe-MOF powders is dominated by the magnetic loss mechanism, and the $tg\delta_\mu$ value is 0.23~0.44. The electromagnetic parameters of composite powders can be regulated by applying an external magnetic field. The electromagnetic parameters of composite powder change sensitively with the strength of the applied magnetic field.

The external regulatory magnetic field has a great influence on the electromagnetic parameters of the FeNi@SrFe-MOF powders. As the strength of the regulated magnetic field increases, the $\varepsilon'$ value decreases significantly, and the $\mu'$ value increases significantly. When the strength of the regulated magnetic field is 472 mT, the Snoke limit frequency of the $\mu'$ value rises to 8.3 GHz.

As the strength of the regulatory magnetic field increases, the $\mu'$ value increases significantly. When the strength of the regulated magnetic field is 472 mT, the $\mu'$ value can break the Snoke limit of the $\mu'$ value, and the Snoke limit frequency of the $\mu'$ value rises to 8.3 GHz.

The FeNi@SrFe-MOF powder coating has good matching characteristics with the air impedance. When the composite powder coating is 4.4 mm thick, the strongest absorption peak, −41 dB, appears at the central frequency of 5.4 GHz, and the effective absorption bandwidth is 2.7 GHz. When the thickness of the composite powder coating is 3.8 mm, the widest effective absorption peak appears in the 5.3 GHz~8.5 GHz frequency band, and the effective absorption bandwidth reaches 3.2 GHz, which shows excellent wave absorption properties.

After magnetic field regulation treatment, the electromagnetic parameters of the FSM powders change significantly, which results in significant changes in the absorption of the coating. When the composite powder coating thickness is 3.5 mm, the central frequency of the strongest absorption peak of the coating changes from 5.4 GHz to 7.8 GHz, the strongest absorption peak of the coating also decreases from −41 db to −59 db, and the coating thickness of the strongest absorption peak decreases from 4.4 mm to 3.5 mm.

**Author Contributions:** Methodology, Z.Y., D.J., L.C. and Z.Z.; Investigation, D.J. and Z.Z.; Data curation, Z.Y., D.J. and L.C.; Writing—original draft, Z.Y. and Z.Z.; Writing—review & editing, Z.Y. and Z.Z.; Visualization, Z.Z.; Supervision, Z.Y.; Funding acquisition, Z.Z. All authors have read and agreed to the published version of the manuscript.

**Funding:** This study was financially supported by the National Natural Science Foundation of China-Research (No. 52061029).

**Institutional Review Board Statement:** Not applicable.

**Informed Consent Statement:** Not applicable.

**Data Availability Statement:** Data are contained within the article.

**Conflicts of Interest:** Lei Chen was employed by the company Jiangxi Hongdu Aviation Industry Group Co., Ltd. The remaining authors declare that the research was conducted in the absence of any commercial or financial relationships that could be construed as a potential conflict of interest.

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
