# Peer review of "The Design and Preparation of Permittivity-Adjustable FeNi@SrFe-MOF Composite Powders"

_coatings, doi:10.3390/coatings14010112_

Round 1
Reviewer 1 Report
Comments and Suggestions for Authors
Comments and remarks to the article:
11. Figure 8 - resize the top row of graphs so that the labels on the ordinate axis are visible (line 136). In addition, for better visual perception, it is advisable to centre the numbering of the figures and replace the labels with numbers, and describe the numbers in the figure caption. The proposed change is provided in the attached file.
22. Lines 153-158. It is not defined what C0 is. Formula (1) has undescribed notations (f, d, σ), and no literature source is provided from which the authors took this formula. On what basis can the authors state that "The C0 of the FSM powders fluctuates with the frequency from 1 to 8.5 GHz"?
33. Line 186, fig. 10. The figures are missing labels a, b, c. The figures should be separated and made the same size.
44. Line 220, fig. 11 d, e, f. In the figure, you need to increase the font size in the axis labels.
55. Line 223, Table 1. In the table, you need to adjust the font size, because some of the captions and numerical values take up two lines.
66. The main part of the article has only one literature reference 17. The authors took the saturation magnetisation for the synthesised materials from this article. The rest of the formulas and explanations are not justified. For example, the conclusion that "the FeNi@SrFe-MOF powder is a kind of efficient absorption material with a magnetic field regulation method" can only be made by comparing it with already known absorbing materials.
Author Response
Reviewer #1:
- Figure 8 - resize the top row of graphs so that the labels on the ordinate axis are visible (line 136). In addition, for better visual perception, it is advisable to centre the numbering of the figures and replace the labels with numbers, and describe the numbers in the figure caption. The proposed change is provided in the attached file.
According to the requirements, the legend has been modified.
- Lines 153-158. It is not defined what C0 is. Formula (1) has undescribed notations (f, d, σ), and no literature source is provided from which the authors took this formula. On what basis can the authors state that "The C0 of the FSM powders fluctuates with the frequency from 1 to 8.5 GHz"?
FSM powders primarily exhibit magnetic loss mechanisms.The eddy current loss effect is the main contributor to magnetic loss mechanisms[18].
|
C0 =(μ"(μ')1/2)/f = 2πμ0σd 2/3 |
(2) |
Where C0 is the eddy current loss effect coefficient, μ0 is the vacuum permeability, f is the electromagnetic wave frequency, σ is the material conductivity, and d is the material thickness. According to this equation, when the material thickness d is determined, the value of C0 is only related to the material conductivity σ, and the value of σ varies little with the frequency f. Therefore, if the magnetic loss is caused by the eddy current effect, C0 will be constant. The C0 of the FSM powders fluctuates with the frequency from 1 to 8.5 GHz. Therefore, the magnetic loss mechanism in the FSM powders includes not only the eddy current loss effect, but also the natural resonance loss.
- Line 186, fig. 10. The figures are missing labels a, b, c. The figures should be separated and made the same size.
I also added labels a, b, and c to the modified text in Figure 10.
- Line 220, fig. 11 d, e, f. In the figure, you need to increase the font size in the axis labels.
The font size in the axis labels in Figures 11 d, e, and f has been modified.
- Line 223, Table 1. In the table, you need to adjust the font size, because some of the captions and numerical values take up two lines.
The font in the table has been modified.
- The main part of the article has only one literature reference 17. The authors took the saturation magnetisation for the synthesised materials from this article. The rest of the formulas and explanations are not justified. For example, the conclusion that "the FeNi@SrFe-MOF powder is a kind of efficient absorption material with a magnetic field regulation method" can only be made by comparing it with already known absorbing materials.
The conclusion "FeNi@SrFe-MOF powder is an efficient absorbing material with a magnetic field modulation method" has been modified, and other content has been modified accordingly, and is identified in yellow in the text.

Reviewer 2 Report
Comments and Suggestions for Authors
This work reports the design and preparation of permittivity adjustable 2FeNi@SrFe-MOF composite powders. The authors need to carefully revise the manuscript for the English language and give a deep interpretation of the results. The following are some recommendations: -
- Abstract: This part is short (has only 5 sentences) and should be rewritten again. Please consider the following in the Abstract;
* The sentence "According to the current requirements of wider band, thinner thickness and lighter weight of absorbing materials, the new kind of FeNi@SrFe-MOF composite powders having the soft magnetic and permanent magnetic characteristics, was constructed, and then the electromagnetic parameters of the powders were adjusted by magnetic field." is long, has many grammar mistakes, and must be modified.
* The sentence "With the increase of the regulatory magnetic field strength, the ε′ of the powders decreased significantly, and the Snoke limit frequency of the μ′ of the powders increased, which breaking the Snoke limit of the μ′ of the powders." has many symbols that their full names must be mentioned before it.
- Introduction: Most of the sentences in this part are long, with which the meaning is lost. The English language must be carefully revised.
- Experiment: This part is well-written and covers all experimental materials and methods. Only, the subsection "2.1. Experimental and Characterization Instruments" should come at the end of that part as "2.3. Experimental and Characterization Instruments" I also recommend that the authors to rename that part as "Experimental".
- The authors did not include the Results and Discussion part, instead, they included two parts named "3. Morphology and structure of composite powders" and "4. Electromagnetic properties and absorption performance of composite powders". Almost all manuscripts must include the Results and Discussion part(s) that report the experimental results and their interpretations. In this part, the authors must mention the full name of any term before its abbreviations. Add a reference number(s) for the equations from (1) to (5). Define all parameters present in all equations.
Conclusions: It is long, not concise, and should be shortened to be concise.
References: The references are not in the journal's house-style format.
Comments on the Quality of English LanguageThe English language must be carefully edited to avoid long sentences, grammar mistakes, and typos.
Author Response
This work reports the design and preparation of permittivity adjustable 2FeNi@SrFe-MOF composite powders. The authors need to carefully revise the manuscript for the English language and give a deep interpretation of the results. The following are some recommendations:
Abstract: This part is short (has only 5 sentences) and should be rewritten again. Please consider the following in the Abstract;
* The sentence "According to the current requirements of wider band, thinner thickness and lighter weight of absorbing materials, the new kind of FeNi@SrFe-MOF composite powders having the soft magnetic and permanent magnetic characteristics, was constructed, and then the electromagnetic parameters of the powders were adjusted by magnetic field." is long, has many grammar mistakes, and must be modified.
This has been corrected in the article and marked in yellow.
* The sentence "With the increase of the regulatory magnetic field strength, the ε′ of the powders decreased significantly, and the Snoke limit frequency of the μ′ of the powders increased, which breaking the Snoke limit of the μ′ of the powders." has many symbols that their full names must be mentioned before it.
- Introduction: Most of the sentences in this part are long, with which the meaning is lost. The English language must be carefully revised.
This has been revised in the article, with many of the symbols adding their full names and identifying them in yellow.
- Experiment: This part is well-written and covers all experimental materials and methods. Only, the subsection "2.1. Experimental and Characterization Instruments" should come at the end of that part as "2.3. Experimental and Characterization Instruments" I also recommend that the authors to rename that part as "Experimental".
The text has been amended accordingly by deleting the heading "2.1. Experimental and Characterization Instruments".
- The authors did not include the Results and Discussion part, instead, they included two parts named "3. Morphology and structure of composite powders" and "4. Electromagnetic properties and absorption performance of composite powders". Almost all manuscripts must include the Results and Discussion part(s) that report the experimental results and their interpretations. In this part, the authors must mention the full name of any term before its abbreviations. Add a reference number(s) for the equations from (1) to (5). Define all parameters present in all equations.
The full names of the terms mentioned in the text have been added and also all parameters in all equations present in the text have been added by definition.
Conclusions: It is long, not concise, and should be shortened to be concise.
The conclusion section has been revised in the article and marked in yellow.
References: The references are not in the journal's house-style format.
The references have been revised to the internal format of the journal.

Reviewer 3 Report
Comments and Suggestions for Authors
The authors submitted "The design and preparation of permittivity adjustable FeNi@SrFe-MOF composite powders: Here are my comments:
1. The full name FeNi@SrFe-MOF should be provided.
2. Both abstract and introduction part should be improved. the novelty of this article is unclear.
3. How about the porosity of these materials?
4. The FTIR of all materials should be provided.
5. SEM data is not sufficient.
6. The scale bar is unclear in some images.
7. Conclusion part is too long.
Comments on the Quality of English LanguageModerate editing of English language required
Author Response
- The authors submitted "The design and preparation of permittivity adjustable FeNi@SrFe-MOF composite powders: Here are my comments:
The full name FeNi@SrFe-MOF should be provided.
The element ratio of Fe and Ni in the FeNi@SrFe-MOF composite powders is 50:50. The powder was prepared as follows: The SrFe12O19 powders with the particle size of about 5μm were dispersed in liquid butylbenzene rubber. The liquid rubber was heated to 150℃ for 12 hours in N2 atmosphere and then was calcined at 300℃ for 2 hours to generate the black SrFe-MOF powders. The powders were dispersed into the aqueous solution of FeSO4·7H2O and NiSO4·6H2O, the temperature was maintained at 85℃ for 1 hour, and the PH value was more than 12. The NH3 gas was continuously generated during the reaction process, and finally the black powders were generated in the water. The final FSM powders were obtained by magnetic separation and drying.
- Both abstract and introduction part should be improved. the novelty of this article is unclear.
The abstract portion of the text has been revised and highlighted in yellow.
- How about the porosity of these materials?
The specific surface area of FSM magnetic powder was 9.16 m2/g. The mesopore pore sizes of FSM magnetic powder were mainly distributed at 9.31 nm, 17.21 nm, and 37.06 nm, and the macroporous pore sizes were mainly distributed at 50.40 nm, 68.50 nm, 86.25 nm, and 117.23 nm, and the total pore volume reached 0.049 cm3/g.
- The FTIR of all materials should be provided.
The strong absorption peak at 584 cm-1 indicates the absorption peak of Fe-O bond vibration or Sr-O bond vibration; 758 cm-1 is the absorption peak of magnetic lattice vibration in strontium ferrite; the absorption peak of the stretching vibration of C-H occurs at 2,868 cm-1; and the absorption peak at 3,702 cm-1 is the stretching vibration associated with the hydrogen on the surface of iron, which suggests that FeNi@SrFe-MOF powders were successfully prepared.
- SEM data is not sufficient.
FTIR data have now been added to the paper to demonstrate the successful preparation of FeNi@SrFe-MOF powders.
- The scale bar is unclear in some images.
The scale of some of the images in the text has been relabeled.
- Conclusion part is too long.
The conclusion section has been revised in the article and marked in yellow.

Round 2
Reviewer 1 Report
Comments and Suggestions for Authors
The corrected version of the article may be published in the journal Coatings. Please review the article again for minor typos. For example, the caption to Fig. 10 starts with a lowercase letter, Figure 11 - FFigure 11. C0 (lines 196 - 204) - the character "0" once as a lowercase index and once as a regular index. Same with Z in lines 263-269. In the list of references, the years of publication of periodicals should be highlighted in bold.
Reviewer 2 Report
Comments and Suggestions for Authors
The manuscript has been revised and greatly improved. I recommend acceptance for publication in the present form.
Comments on the Quality of English LanguageThe English language has been revised and improved.
Reviewer 3 Report
Comments and Suggestions for Authors
this manuscript should be accepted in this journal without further review.